# Neural Networks Application for Accurate Retina Vessel Segmentation from OCT Fundus Reconstruction

**DOI:** 10.3390/s23041870

**Published:** 2023-02-07

**Authors:** Tomasz Marciniak, Agnieszka Stankiewicz, Przemyslaw Zaradzki

**Affiliations:** Division of Electronic Systems and Signal Processing, Institute of Automatic Control and Robotics, Poznan University of Technology, 60-965 Poznan, Poland

**Keywords:** biometrics, retina vessel segmentation, convolutional neural networks, UNet, optical coherence tomography, fundus reconstruction

## Abstract

The use of neural networks for retinal vessel segmentation has gained significant attention in recent years. Most of the research related to the segmentation of retinal blood vessels is based on fundus images. In this study, we examine five neural network architectures to accurately segment vessels in fundus images reconstructed from 3D OCT scan data. OCT-based fundus reconstructions are of much lower quality compared to color fundus photographs due to noise and lower and disproportionate resolutions. The fundus image reconstruction process was performed based on the segmentation of the retinal layers in B-scans. Three reconstruction variants were proposed, which were then used in the process of detecting blood vessels using neural networks. We evaluated performance using a custom dataset of 24 3D OCT scans (with manual annotations performed by an ophthalmologist) using 6-fold cross-validation and demonstrated segmentation accuracy up to 98%. Our results indicate that the use of neural networks is a promising approach to segmenting the retinal vessel from a properly reconstructed fundus.

## 1. Introduction

Fundus photography (2D) is a retinal imaging technique that is commonly used in ophthalmology. It allows for diagnosing and monitoring the course of eye diseases, such as retinopathy, glaucoma, or senile macular degeneration. Segmentation of vessels based on the fundus image is also frequently performed to align multiple retinal images, e.g., acquired at various points (such as the macula and the head of the optic nerve), during multiple clinical visits, with different devices or even with different imaging modalities [1]. Fundus photography can also be used in advanced biometric identification systems [2,3].

Another device that is used for imaging the retina is the optical coherent tomography (OCT) device, which enables the obtaining of 3D sections and, thus, in-depth observation of individual layers of the retina. The so-called B-scans can be used to reconstruct the image of the fundus. An advantage of using OCT devices is less invasiveness compared to the fundus camera, i.e., no strong illumination (flash) at the time of image acquisition.

Furthermore, analyzing retina biometric characteristics in 3D is useful for also assessing eye diseases affecting the vessels directly [4]. A comprehensive 3D vessel structure can also be beneficial for increased accuracy in identifying individuals in a biometric security system.

The fundamental difference between the fundus image and the reconstruction obtained from OCT scans is its resolution. Fundus images have the same resolution in both directions. The OCT reconstruction has a different resolution in the vertical and horizontal directions. Detailed information can be found in Section 1.1. In the case of fundus reconstruction based on OCT scans, an important processing step is the proper selection of a range of retinal layers. Additionally, the automatic segmentation of vessels is not commonly available since algorithms have to overcome image processing problems such as speckle noise, low/uneven data resolution and contrast, and crossing of vessels in the axial direction [5,6,7].

The main contributions of the authors presented in the article are:Reconstruction selection based on segmentation in B-scans,Preparation of a special dataset based on 3D OCT scans,Evaluation of the effectiveness of vessel segmentation using various neural networks.

### 1.1. Related Works

As in the case of general-purpose segmentation methods, retinal vessel segmentation methods can be divided into classical image processing solutions and those that use artificial intelligence. In addition, it should be noted that individual solutions are related to the type of acquisition. Therefore, solutions for the image from the fundus camera and reconstruction based on 3D OCT scans will be discussed separately.

#### 1.1.1. Vessels Segmentation from Fundus Images

Experimental studies of vessel segmentation based on fundus images are conducted using several datasets. As mentioned earlier, the fundus camera images take advantage of the high contrast and resolution of the image. The most popular datasets with manual reference segmentation (so-called ground truth) are:DRIVE (Digital Retinal Images for Vessel Extraction)—40 images (divided into 20 training images and 20 test images) digitized to 584 × 565 color pixels and saved in tiff format [8,9],STARE (Structured Analysis of the Retina)—20 retinal slides digitized to 650 × 500 color pixels in portable pixmap (PPM) format [10],CHASE_DB1 (Child Heart and Health Study)—28 images digitized to 999 × 960 color pixels in jpg format [11].

A review of various solutions was performed in [12]. The segmentation process can be realized with the use of two strategies:Unsupervised—for example, line detectors, co-occurrence matrix, thresholding, difference-of-Gaussian filters,Supervised—where ground truth is required, and after feature extraction, machine learning classifiers such as a nearest-neighbor classifier, a Bayesian, or a Gaussian Mixture Model are used.

The results of various segmentation methods for the DRIVE and STARE databases presented in paper [12] show that, for unsupervised methods, the specificity results are obtained at a level of up to 0.9750, and in the case of supervised methods, the specificity even reached 0.9819.

Blood vessel segmentation can be performed using currently popular deep neural networks. Such networks require the preparation of a large number of samples (the so-called patches with dimensions of, for example, 27 × 27 pixels) and are additionally preprocessed with global contrast normalization, zero-phase whitening, and augmented using geometric transformations and gamma corrections. The effectiveness of deep neural network solutions can be assessed using the AUC (area under the receiver operating characteristic (ROC) curve), which reaches values above 0.9 [13]. The IterNet [14] solution, based on UNet models for DRIVE, STARE, and CHASE-DB1 datasets, obtain the AUC values of 0.9816, 0.9881, and 0.9851, respectively. In the case of the DRIVE database [9], a challenge is organized, in which the segmentation quality is assessed using the DICE coefficient, currently reaching the value of 0.9755.

#### 1.1.2. Vessels Segmentation from OCT Images

Segmentation of blood vessels based on OCT scans is currently not the subject of numerous studies compared to the previously discussed segmentation based on images from the fundus camera. Due to the specificity of image acquisition and the need for preprocessing, direct application of the methods used for the camera fundus is ineffective. Another drawback is the lack of databases of OCT fundus reconstruction images with manual reference segmentations. Solutions dealing with the segmentation of retinal vessels based on OCT can be divided into three groups [15]:unimodal—3D OCT only based methods,multimodal—hybrid methods that also use, in addition to 3D OCT data, data from the fundus camera or scanning laser ophthalmology,optical coherence tomography angiography (OCTA)—a solution that can be found in the newest OCT devices.

In our research, we focus on using only 3D OCT measurement data, which are available on most devices. Therefore, the state of research for such methods is presented below.

The topic of segmentation of blood vessels from eye fundus reconstruction based on OCT was presented in 2008 in a publication prepared by M. Niemeijer et al. [16]. Data acquisition was performed using Zeiss Meditec Cirrus OCT, and 15 optic nerve head-centered spectral 3D OCT scans of 15 normal subjects were used for the automatic segmentation experiment. The method is based on a supervised pixel classification of a 2-D projection and consists of the following steps: layer segmentation, vessel image projection, and pixel classification using the kNN classifier. Depending on the type of projection: naive (i.e., simple A-scans averaging) or smart (averaging from segmented B-scans), the area under the curve is 0.939 and 0.970, respectively. It should be noted that this solution applied to the segmentation of Nerve Canal Opening (NCO) gives an AUC value of 0.79 [1].

An important issue in the classification of a given pixel to the class of blood vessels is the selection of the appropriate threshold based on probability maps. A typical approach is to take the threshold as 0.5. As in the case of fundus image analysis [17], a dual-threshold iterative algorithm (DTI) can be used, which results from taking into account the surroundings of a given pixel. The threshold can also be performed using different techniques [18], e.g., slope difference distribution (SDD) clustering and threshold selection, which have been proposed for magnetic resonance (MR) images and is effective for the detection of ventricle ROI.

Another important element of the process of detecting blood vessels is the correct segmentation of the layers and then their selection as borders in the stage of preparing the reconstruction image [15]. The so-called shadowgraph can be computed for different ranges. Taking into account intensities of tissue reflectance in the GCL layers allows for emphasizing the vessels that exist in the superficial vascular complex. After applying the smoothing filter, detection (binarization) can be performed using the threshold function. The results of the accuracy obtained reached the value of 94.8% with a precision of 75.2%.

Segmentation can also be performed using only filter operations. In article [19], the vessel enhancement filter is applied to detect tubular geometric structures and suppress remaining noise and background. A vesselness measure is obtained on the basis of all eigenvalues of the Hessian [20]. The filter-based method is also applied in [21], but in order to obtain a high-quality projection image, the authors adopted histogram equalization followed by Wiener filtering. Filtering can also be used in conjunction with morphological operations, resulting in a precision of 83.9% [22].

Effective and accurate 3D registration methods for retinal SD-OCT images are also being explored in [21] based on [19]. Such solutions require the use of two three-dimensional OCT scans, which, after appropriate labeling, allow one to obtain the *x-y* direction registration and the *z* direction registration. Three-dimensional segmentation is also analyzed for parameters such as the correction of crossings or bifurcations [23].

In conclusion, it can be noted that, to the best knowledge of the authors of this article, in the case of images of the fundus of the human eye reconstructed on the basis of OCT, there is no comprehensive study of the use of convolutional neural networks to detect retinal blood vessels.

## 2. Materials

### 2.1. OCT Image Dataset

As was mentioned in the previous section, there are no publicly available datasets of fundus reconstruction images gathered with the use of OCT. The accessible OCT datasets focus on retina cross-sections necessary for research in detecting retina pathologies and improving automatic retina layer segmentation. Furthermore, most of them include only a single cross-section for one patient, which is insufficient to provide an en face reconstruction of the macula region. Thus, to train a neural network in the task of retinal blood vessel segmentation from OCT, it was necessary to gather a set of OCT scans within our public CAVRI (Computer Analysis of VitreoRetinal Interface) dataset [24]. The subset of 3D OCT macula scans with manual reference annotations utilized for this research is called CAVRI-C. It is a collection of 24 scans acquired with the Avanti RTvue OCT (Optovue Inc., Fremont, CA, USA). The images were obtained from 12 healthy volunteers (left and right eye) with an average age of 27 years. The resulting scans have a resolution of 141 × 385 × 640 px. The data represents 7 × 7 × 2 mm of tissue, which gives an axial resolution of 3.1 μm and a transversal resolution of 18.2 μm.

The obtained 3D scans were utilized for the reconstruction of fundus images with the methodology described in Section 2.2. Manual segmentation of 9 retina layers borders (ILM, NFL/GCL, GCL/IPL, IPL/INL, INL/OPL, OPL/ONL, ONL/IS, IS/OS, and RPE/CHR), as well as blood vessels (from the fundus reconstruction image), was carried out by a team of experts from the Department of Ophthalmology, Chair of Ophthalmology and Optometry, Heliodor Swiecicki University Hospital, Poznan University of Medical Sciences. For the layers annotation, a custom-made public software OCTAnnotate was used [25].

### 2.2. Fundus Reconstruction from 3D OCT Scan

It is possible to reconstruct a fundus image from a 3D OCT scan, although it is not a straightforward process, and accurate reconstruction can be challenging. The basic idea of this process is to average OCT data in the vertical (axial) direction (the so-called A-scans), as illustrated in Figure 1. As can be found in the literature, the early approaches used a whole 3D scan when averaging each A-scan [26]. The disadvantage of this approach is the inclusion of irrelevant portions of the scan (above and below the retina), as well as noise.

A better approach utilizes selected layers of the retina from the OCT scan to take advantage of the contrast between vessel shadows and the hyper-reflective tissue of the outer retina (namely OS and RPE layers) [1,16,27]. The accuracy of the reconstructed image will depend on the quality of the OCT data and the algorithm used to select relevant retina tissue [15,22].

It should be noted that the resolution of the reconstructed fundus image depends on the scanning protocol of the volumetric OCT data. Typically, a reconstructed fundus image (acquired with the fixed scanning OCT parameters) has uneven resolutions for the fast and non-fast scanning directions. In our experiment, the 3D Retina scanning protocol, employed by the Avanti RTvue device (Optovue Inc., Fremont, CA, USA) [28], consists of 141 B-scans of 385 pixels width, representing a 7 × 7 mm retina area. To obtain the vessel map corresponding to their real geometric structure, we used bicubic interpolation.

During the experiments, we tested three combinations of retina regions for the fundus image reconstruction. The regions utilized for analysis include:ganglion cell layer (GCL)inner plexiform layer (IPL)outer segment of photoreceptors and retina pigment epithelium layer (OS+RPE).

Figure 2 illustrates an example of an acquired OCT B-scan image with manual annotation of the borders for the above-listed layers. Two superficial vessels are visible as bright areas in the GCL layer (between the yellow lines illustrating NFL/GCL and GCL/IPL borders). Dark regions underneath, visible in the OS+RPE section (between the green lines of IS/OS and RPE/CHR borders) are the shadows of these vessels. The red line drawn in the B-scan image represents the lower boundary of the IPL layers, namely IPL/INL.

Let I(x,y,z) represent the volumetric OCT data in the spatial coordinate system, in which *x* and *y* represent pixel indexes in the horizontal fast and non-fast scanning directions, respectively, and *z* stands for pixel index in the axial direction. Using this definition, we specify the following retina layers projections:*GCL layer projection*—a projection of the layer defined between the NFL/GCL and GCL/IPL borders (illustrated by yellow lines in Figure 2). The pixels between the specified layer borders are averaged along the *z*-axis with the following equation:
(1)$$P_{GCL}\left(x,y\right)=\frac{\sum_{z=L_{GCL/IPL}\left(x,y\right)}^{L_{NFL/GCL}\left(x,y\right)}I\left(x,y,z\right)}{L_{NFL/GCL}\left(x,y\right)-L_{GCL/IPL}\left(x,y\right)}$$
where $L_{NFL/GCL}$ denotes the vertical border location between the NFL and GCL for *x* and *y* coordinates, and $L_{GCL/IPL}$ denotes the border location between the GCL and IPL layers.*GCL+IPL layers projection*—a projection of a region encompassing two neighboring layers defined between the NFL/GCL and IPL/INL borders (see Figure 2). Similarly to before, the pixel values are averaged along the *z*-axis:
(2)$$P_{GCL+IPL}\left(x,y\right)=\frac{\sum_{z=L_{IPL/INL}\left(x,y\right)}^{L_{NFL/GCL}\left(x,y\right)}I\left(x,y,z\right)}{L_{NFL/GCL}\left(x,y\right)-L_{IPL/INL}\left(x,y\right)}$$
where $L_{IPL/INL}$ denotes the vertical border location between the IPL and INL for *x* and *y* coordinates.*OS+RPE layers projection*—a mean of pixels intensity values in each A-scan from the area of hyper-reflective tissues, i.e., OS and RPE layers (confined between the green lines in Figure 2):
(3)$$P_{OS+RPE}\left(x,y\right)=\frac{\sum_{z=L_{IS/OS}\left(x,y\right)}^{L_{RPE/CHR}\left(x,y\right)}I\left(x,y,z\right)}{L_{RPE/CHR}\left(x,y\right)-L_{IS/OS}\left(x,y\right)}$$
where $L_{IS/OS}$ denotes the border location between the inner and outer segments of photoreceptors, $L_{RPE/CHR}$ denotes the border location between the RPE and choroid layers.

Figure 3 illustrates these layers’ projections. It is worth noticing that in the GCL projection, the vessels appear brighter than the surrounding tissue (see Figure 3a,b), while in the OS+PRE projection (Figure 3c), the situation is reversed. It can also be observed that the GCL layer allows thin vessels present in the superficial vascular complex that is too small to leave a significant shadow trace in the outer layers (i.e., IS/OS) to be visualized.

Combining projections defined by Equations (Equation 1)–(Equation 3) allows the enhancing the retina vessel network further. Some vessels span across these two layers, thus we consider both GCL and IPL layers for vessel projection. Additionally, the further the fovea, the deeper they penetrate retina layers (and move from GCL to the IPL layer).

Reconstruction P1—is a standard outer retina projection calculated from only OS+RPE layers, as described in Equation (Equation 3). This reconstruction, proposed in [1], is frequently used in OCT-based retina vessel research. An example of this method is illustrated in Figure 4a.Reconstruction P2—is calculated as a function of weighted projections of GCL and OS+RPE layers as described by Equation (Equation 4):
(4)$$P_{2}\left(x,y\right)=w_{1}P_{OS+RPE}\left(x,y\right)+w_{2}P_{GCL}\left(x,y\right)$$
where parameters $w_{1}$ and $w_{2}$ are used to weigh the influence of vessels and their shadows, the OS+RPE projection describes Equation (Equation 3), and the GCL projection is calculated with Equation (Equation 1). Initial experiments led to the values of $w_{1}=1.7$ and $w_{2}=0.8$ selected empirically.(3)Reconstruction P3—is similar to P2, but instead of using GCL projection, it utilizes the projection of both GCL+IPL layers, as can be seen in Equation (Equation 5):
(5)$$P_{3}\left(x,y\right)=w_{1}P_{OS+RPE}\left(x,y\right)+w_{2}P_{GCL+IPL}\left(x,y\right)$$Here, the values of weighing parameters are $w_{1}=2$ and $w_{2}=1.2$, also chosen experimentally.

Figure 4 shows an example of the $P_{2}$ and $P_{3}$ reconstruction images with the enhanced contrast of small vessels. These images were obtained from the 3D OCT data. It can be seen that the tissue reflectance in the GCL layer allows for emphasizing vessels that exist in the superficial vascular complex and are too small to leave a significant shadow trace.

The extraction of retina layers and subsequent projection calculation was performed in a Matlab/Simulink environment [29] on raw data exported from the OCT device.

## 3. Methods

In the course of this research, we evaluated five convolutional neural networks for the semantic segmentation of retina vessels from OCT-reconstructed fundus images. The task of the network is to assign each pixel of the image as either a vessel or non-vessel, making this a binary classification problem. The following network architectures were tested: UNet, IterNet, BCDU-Net, SA-UNet, and FR-UNet. Their short description can be found below. Figure 5 illustrates the general overflow of the proposed approach.

### 3.1. UNet

The U-Net architecture [30] is based on a fully convolutional network. It consists of a series of convolution layers that process input data through two consecutive symmetrical parts: contracting and expansive, giving the network a u-shaped structure. The contracting path is a typical convolutional network consisting of 4 levels of repeated 3 × 3 convolutions, followed by a rectified linear unit (ReLU) and a max pooling operation. This process reduces the spatial information while increasing the feature information. After a final pooling, the last (5th) level also consists of repeated 3 × 3 convolutions before being subjected to the expansive part, in which the pooling is replaced with upsampling operators to increase the layers’ size back to the original resolution. A large number of feature channels and concatenation operations between the contracting and expanding layers allow the network to pass context information to higher-resolution layers and learn a detailed output.

We selected the UNet structure due to its very good performance in various segmentation tasks, especially in medical applications. In our application, the input to the network is a 1-channel OCT-reconstructed fundus image. The network outputs a 2-channel (vessel/non-vessel) probability map of the same size as input, containing the information to which class each pixel should be classified.

### 3.2. IterNet

The second utilized network, namely the IterNet [14], is an extension of the previously described UNet architecture. It is constructed from a basic (5-level) UNet module, supplemented with several (in our case, 3) refinery mini-UNet modules. The mini-UNet module has 1 level less than a basic UNet (i.e., 4 levels). The second-last level of the first module is treated as an input to the next module, and a similar procedure is repeated for the following modules. Each module is trained to fit the correct segmentation labels with an individual loss function. Thus the refinery modules learn immunity to false or missing vessel patterns. An additional feature that allows avoiding the overfitting problem is additional skip-connections between the modules and weight-sharing (i.e., all mini-UNet share the same weights and biases), which also reduces the number of necessary training samples.

The IterNet structure was designed to learn the human vessel network structure from 128 × 128 px patches of color fundus photographs. Its strength is in the ability to gradually connect split micro-vessels during the iterative prediction of the refinery modules. In our experiment, we aim to take advantage of this ability and subject patches of our fundus reconstruction images (extrapolated to 3 channels) as input to the IterNet. The network outputs a 1-channel probability map for each pixel with vessels of the same size as the input image.

### 3.3. BCDU-Net

The next promising network is the Bi-Directional ConvLSTM UNet (BCDU-Net) [31]. This method supplements a 4-level UNet structure with bidirectional convolutional LSTM layers to take advantage of both semantic and high-resolution information in a non-linear way. Additionally, it incorporates densely connected convolution layers to include collective knowledge in the representation. Furthermore, batch normalization is employed to improve the convergence rate.

This network showed advantageous results for three different types of medical segmentation applications: retina blood vessels, skin lesions, and lungs. Following the authors’ solution, we employed this network for reconstructed fundus images utilizing 64 × 64 px patches as input images to train the network. The network outputs a 1-channel matrix of the size of the input with probability information for each pixel belonging to a vessel.

### 3.4. SA-UNet

Another tested architecture was the SA-UNet [32]. Here, the authors introduced a spatial attention module in a 4-level U-Net structure between the encoder and decoder paths. The main idea of this solution is to exploit the attention feature (that well retains structural information) from various complex network models in a lightweight, better interpretable and comparable (with respect to accuracy) model.

This network was designed to effectively perform detailed segmentation of fine retina vessels from color fundus images. We hope to take advantage of this solution by subjecting our reconstructed fundus images (with 3-channels) to this architecture. The resulting prediction is a 1-channel probability map.

### 3.5. FR-UNet

The fifth neural network model considered in our application is a new approach called Full-Resolution and Dual-Threshold Iteration based on the UNet architecture [17]. It extends the original approach by horizontal and vertical expansion through a multiresolution convolution interactive mechanism. The shallow stage provides more refined semantic information, and the deep stages increase the local receptive field. In contrast to traditional encoder-decoder architecture, the first stage of FR-UNet continuously integrates high-level contextual information while maintaining the original resolution. The FR-UNet additionally incorporates a feature aggregation module that integrates multiscale feature maps from adjacent stages. The authors also proposed a dual-threshold iterative algorithm to improve vessel connectivity.

This novel and straightforward approach aims to alleviate the problem of losing spatial information important in the segmentation of thin vessels with low contrast. Although it was developed for color fundus photographs, we hope to show its advantages in the application of OCT-reconstructed fundus images.

## 4. Results

### 4.1. Experiment Setup and Evaluation Metrics

To examine the effectiveness of vessel segmentation based on fundus reconstructions from OCT data, we conducted a series of experiments with the above-described networks. The quality of the predictions obtained was compared with the manual segmentations (experts) using basic metrics, such as the area under the ROC curve (AUC), accuracy, sensitivity, precision, specificity, and F1-score [33]:(6)$$\mathrm{Accuracy}=\frac{\mathrm{TP}+\mathrm{TN}}{\mathrm{TP}+\mathrm{TN}+\mathrm{FP}+\mathrm{FN}}$$
(7)$$\mathrm{Sensitivity}=\frac{\mathrm{TP}}{\mathrm{TP}+\mathrm{FN}}$$
(8)$$\mathrm{Specificity}=\frac{\mathrm{TN}}{\mathrm{FP}+\mathrm{TN}}$$
(9)$$\mathrm{Precision}=\frac{\mathrm{TP}}{\mathrm{TP}+\mathrm{FP}}$$
(10)$$\mathrm{F1-score}=\frac{2*\mathrm{TP}}{2*\mathrm{TP}+\mathrm{FP}+\mathrm{FN}}$$
where TP is true positive, TN—true negative, FP—false positive, and FN—false negative. We used the implementation of these equations submitted with the experiment code provided by the authors of the UNet architecture.

For each of the neural networks, six-fold cross-validation was carried out, in which 20 fundus images were used for training and the remaining 4 for the test. The 20 training images were further split into validation and training subsets with ratios depending on the original recommendations of the software’s authors.

During all experiments, we used official IterNet, BCDU-Net, SA-UNet, and FR-UNet implementations shared by their authors on GitHub. For UNet, we used Daniele Cortinovis’s implementation [34]. While adapting the tested neural networks to process reconstructed fundus images, we tried to change the original code as little as possible. The links to the code are available in the Data Availability Section at the end of the article.

The experiments were carried out on Google Colab and Paperspace. The original neural network code is written in Python 2. We have rewritten it to version 3 and changed the way training and validation accuracy is calculated by using only the second output channel (to correspond to the way the test accuracy is measured). UNet, IterNet, and SA-UNet were trained using a Nvidia M4000 graphic processing unit (GPU). The average time for one fold was 60, 7, and 25.5 min, respectively. BCDU-Net was trained on the Nvidia A100 GPU, with a 6.5-min average time for one fold. The training of the FR-UNet network was carried out using a Nvidia Tesla T4 GPU, on which the average time of one training was about 25 min.

### 4.2. Preprocessing and Data Augmentation

Data preprocessing and augmentation were carried out using the same methods that were used by the authors of the tested neural networks. In the case of UNet and BCDU-Net, the fundus images were subjected to the same preprocessing, including conversion to grayscale, z-score normalization, CLAHE histogram equalization (Contrast Limited Adaptive Histogram Equalization), and gamma correction. However, we omitted the stage of cropping the edges of the images occurring in the original implementation. No data augmentation was performed.

Images processed by the IterNet network were not preprocessed, while the augmentation process consisted of random changes in brightness, contrast, saturation, and geometric operations, such as random rotation (±20 deg), shearing, reflection, shifting, and zooming (in the range of 〈0.8, 1.0〉).

The augmentation of the images used during SA-UNet training included randomly changing the color balance, brightness, contrast, sharpness, adding Gaussian noise, performing random rotation and cropping. For FR-UNet, the images were z-score normalized, randomly rotated by 0, 90, 180, or 270 degrees, and randomly flipped vertically and horizontally.

Table 1 lists the setup parameters for the tested network architectures.

### 4.3. Vessels Segmentation with UNet

This subsection presents a quantitative comparison of vessel segmentation with a UNet architecture using three reconstructions, P1, P2 and P3, described in Section 2.2.

Figure 6, (a), (c), and (e), present the loss values used to evaluate the training for P1, P2, and P3, respectively. The validation process for those reconstructions is presented in plots (b), (d), and (f), respectively.

For most experiments, the final training loss drops to around 0.1. However, we can notice in the case of P2 and P3 reconstructions that there are outliers in which the training loss function is noticeably larger (0.16 for P2 and 0.15 for P3). The lowest validation loss values of around 0.1 are achieved with P1 reconstruction. For the P2 and P3 reconstructions, the validation loss is higher (between 0.14 and 0.2), and we also observe greater variation between individual folds.

The accuracy plots for the training subsets using P1, P2, and P3 reconstructions are presented in Figure 7—(a), (c), and (e), respectively. The validation process for those reconstructions is illustrated in subplots (b), (d), and (f), respectively.

The training accuracy in all experiments reaches values equal to approximately 96%. The greatest validation accuracy is obtained with P1 reconstruction. Again, the P2 and P3 reconstructions lead to worse results and a larger spread between folds.

### 4.4. Vessel Segmentation with IterNet

Figure 8 shows the loss and accuracy plots obtained during the training of the IterNet network. In all experiments, the network was convergent. For all reconstruction types, the final training loss falls below 0.4, and the training accuracy exceeds 97%. In most cross-validations, the learning curves are similar; only for the third fold using the P1 reconstruction an outlier occurs.

### 4.5. Vessels Segmentation with BCDU-Net

The loss and accuracy plots obtained with the BCDU-Net architecture are shown in Figure 9 and Figure 10, respectively. According to loss plots for all three reconstructions (see Figure 9a,c,e), the learning process stabilizes from the 15th epoch—training loss reaches a minimum value of around 0.03–0.04 (depending on the cross-validation subset). On the other hand, it can be noticed from the validation loss plots (see Figure 9b,d,f) that the optimal learning point was achieved after around the eighth epoch, after which the validation loss starts to increase, indicating overtraining of the network.

It can also be observed that, apart from a few individual outlier values for validation loss using the P1 and P2 reconstructions, all cross-validation samples present similar results with regard to the obtained loss values during training and validation.

Analyzing Figure 10, it can be noticed that after the initial rapid increase in accuracy value in the first two epochs, the accuracy for the training gradually increases to saturate after the 15th epoch on the value of 0.985–0.99 for all three reconstructions (see Figure 10a,c,e). Similar, although slightly lower values are presented for validation accuracy (in Figure 10b,d,f), where the accuracy saturates after the eighth epoch, on average, to 0.98, 0.976, and 0.978 for P1, P2, and P3, respectively. The accuracy plots also indicate concurrence for all cross-section results.

### 4.6. Vessels Segmentation with SA-UNet

Figure 11 shows the training and validation loss of the SA-UNet. In all experiments, the network is convergent, and the training and validation loss drops to around 0.175. From the validation loss plots, it can be noticed that after the 60th epoch, the network is trained, and no further improvement in the loss value occurs. No significant differences between fundus reconstruction methods can be observed.

The accuracy plots for SA-UNet architecture are presented in Figure 12. The obtained accuracy value for both training and validation is about 96% for all three types of fundus reconstructions. Interestingly, the validation accuracy does not change for the first 15 epochs before increasing from 0.925 to 0.96. This phenomenon was also present in the experiment performed by the authors of the SA-UNet architecture [32].

### 4.7. Vessels Segmentation with FR-UNet

The loss plots for FR-UNet are presented in Figure 13. In the case of training, it can be observed that for each of the three reconstructions, P1, P2 and P3, the curves for each of the cross-validations are very similar. The learning process stabilizes after 30 epochs and reaches 0.05.

Figure 14 shows the accuracy plots of the FR-UNet. The accuracy plots for training also show very similar results for all six folds. This observation is evident for all types of reconstructions (P1, P2, P3). The training saturates after 35 epochs at a value of 0.975. Similar results are found for the validation accuracy, where the accuracy after 35 epochs achieves 0.982, 0.982, and 0.985 for P1, P2, and P3, respectively.

### 4.8. Comparison of Models and Fundus Reconstruction Methods

This section presents qualitative and quantitative comparisons of fundus segmentation results obtained with the tested neural network models. The averaged values of the metrics obtained (accuracy, sensitivity, specificity, precision, F1-score, and AUC) are listed in Table 2. In this table, the best results for a given neural network are marked in bold, and additionally, an asterisk indicates which reconstruction, together with the particular network, gives the overall best value for each metric. For comparative purposes, Table 2 also contains results for traditional solutions based on shadowgraphs [15] and morphological operations [22].

It can be observed that depending on the reconstructions, the best results are obtained for solutions P1 and P3, regardless of the type of neural network used. The proposed reconstruction P3 gives the best results in most cases.

The FR-UNet network, which was originally proposed for the detection of blood vessels in color fundus photographs, obtains the highest values for selected metrics in the case of OCT-reconstructed fundus. For this neural network, the impact of the dual-threshold iterative (DTI) algorithm [17] was also tested. The accuracy, sensitivity, F1 score, and AUC for FR-UNet are the highest (for the P3 reconstruction). The use of DTI in the case of FR-UNet improves the sensitivity, but reduces the values of other metrics, especially the precision. The UNet network, which was the basis of the other network architectures, has significantly lower metric values. Both UNet and FR-UNet networks achieve the highest sensitivity value among all networks, and this is the case with the reconstruction P1 for UNet and reconstruction P3 for FR-UNet.

BCDU-Net achieved results similar to IterNet, and the distribution of the best results is identical to IterNet, with a predominance of the highest values for the P3 reconstruction. Accuracy and AUC are the same as for IterNet, while the difference for sensitivity and F1-score is only 0.007 and 0.002, respectively (also for P3). IterNet (using P1) outruns BCDU-Net and FR-UNet with regard to specificity by only 0.002 and 0.005, respectively, but precision by a much greater difference of 0.023 and 0.055, respectively.

In the case of the SA-UNet network, it can be observed that all metrics are the best for the proposed P3 reconstruction, but they are slightly lower compared to the metrics for P3 of the IterNet and BDCU-Net networks. The obtained F1-score (for all networks) is comparable to the results reported in the literature that use color fundus images.

It should be added that the classical methods can achieve good results with respect to accuracy and specificity measures (around 90–98%). Nevertheless, for sensitivity, precision, and F1-score, the obtained values are much lower compared to the solution using neural networks. Only in the case of morphological filtering—BML approach is the F1-score 0.816, but this value is lower in comparison to FR-UNet (0.857).

Figure 15 illustrates examples of segmentation predictions using the five described neural networks and the proposed reconstructions P1, P2, and P3. The green color represents proper vessel segmentation, red indicates a lack of segmentation for a vessel marked by an expert, and blue denotes pixels classified as vessels that are not in the ground truth. For comparison, manual segmentation is also included in the figure.

It can be observed that the UNet architecture provides thicker vessels with many artifacts resulting from a lack of continuity in the vessel path. The network that obtained the best metric values, i.e., IterNet, simultaneously provides more details and better continuity, especially utilizing the P3 reconstruction. The best qualitative results for thin vessels can be observed using BCDU-Net. This architecture also provides a very detailed and continuous vessel network. Interestingly, SA-UNet does not provide the segmentation of fine vessels, which is especially visible for the reconstruction P1.

In general, the P1 and P3 reconstructions allow for better continuity of the vessel network. For the P2 reconstruction, many vessels are broken or simply not present. This qualitative analysis in Figure 15 supports the quantitative data presented in Table 2.

## 5. Discussion

In general, it can be seen that all neural networks achieve AUC values greater than 0.957, and in the case of FR-UNet, the AUC parameter is 0.991 for reconstruction P3. The analysis of parameters, such as accuracy, sensitivity, F1-score and AUC, indicates that the FR-UNet architecture works best for the prepared dataset. Nevertheless, slightly lower results are given by IterNet and BCDU-Net. It should also be mentioned that neural networks allow for obtaining better precision (by about 12–16%), sensitivity (by about 34%), and F1-score (by about 32%) values compared to the classic shadowgraph-based approach [15].

Preparing the reconstruction of the fundus image based on 3D optical coherence tomography scans requires careful conduct of the initial stage consisting of the correct segmentation of the layers in individual B-scans of the three-dimensional OCT scan. The accuracy of delineating the boundaries of the retinal layers on B-scans affects the fundus reconstruction process. As experiments have shown, it is difficult to clearly indicate which reconstruction method is the best, although P3 gives very promising results, while P2 is the worst. It turns out that no reconstruction can be explicitly selected as appropriate for a given neural network to provide the best results for all evaluation parameters. This is not the case, however, for the SA-UNet network, where the P3 reconstruction obtained the best results for all assessment criteria. Additionally, for this reconstruction (P3), regardless of the network type, such parameters as accuracy or AUC are always the highest.

Although the results of the presented algorithm are good, several issues remain. First, we have not tested the method on OCT scans of the optic nerve head. However, since the vasculature around the NCO is much thicker than the vessels network in the macula, the performance of the segmentation algorithm could be even better. As the results in Figure 15 show, the detection of the wider vessels is generally good. Second, the same images show the problem of fragmentation or lack of proper segmentation for the thinner vasculature. The main cause of this problem is the lower contrast between the thin vessels and the surrounding tissue, as well as the very low resolution of the image (the thin vessels have a width of 1 px). In future work, we hope to improve the proposed method not only to provide an unbroken vessel network map but also to test the area around the NCO.

## 6. Conclusions

In this article, we present an approach to using neural networks for retina vessel segmentation from fundus images reconstructed from a series of OCT cross-sections. The experimental studies presented have shown that various neural networks can be used to effectively implement blood vessel detection. The prepared database of 3D OCT scans with fundus reconstructions using three types of reconstruction covers the area of the macula. This area is of particular importance in the treatment of degenerative retinal changes. Proper visualization of blood vessels allows supporting diagnostic procedures to monitor the progression of the disorder and select case-appropriate treatment algorithms. Ophthalmologists can serve as the basis for further studies on predictions of anatomical success with medical or surgical intervention. In addition, they can provide guidance for the safe positioning of surgical instruments in vitrectomy surgery.

## Figures and Tables

**Figure 1 sensors-23-01870-f001:**
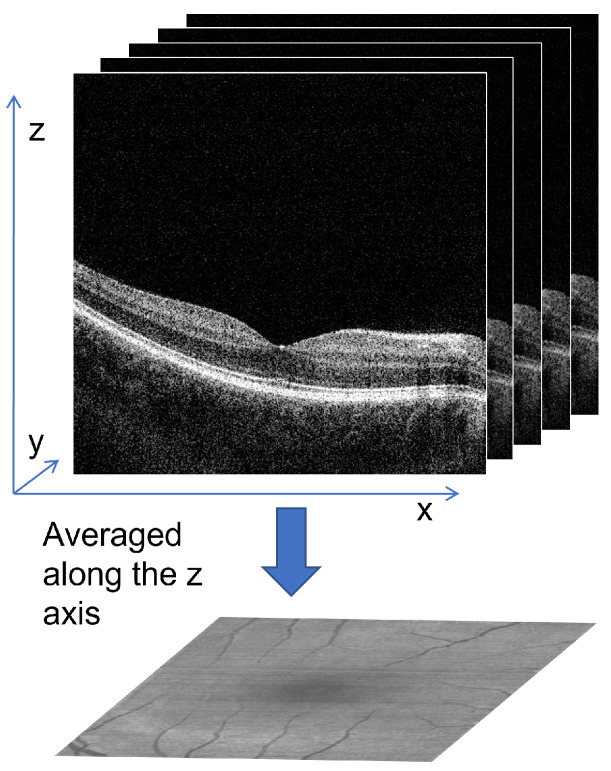
Illustration of simple projection generation from a whole 3D OCT scan.

**Figure 2 sensors-23-01870-f002:**
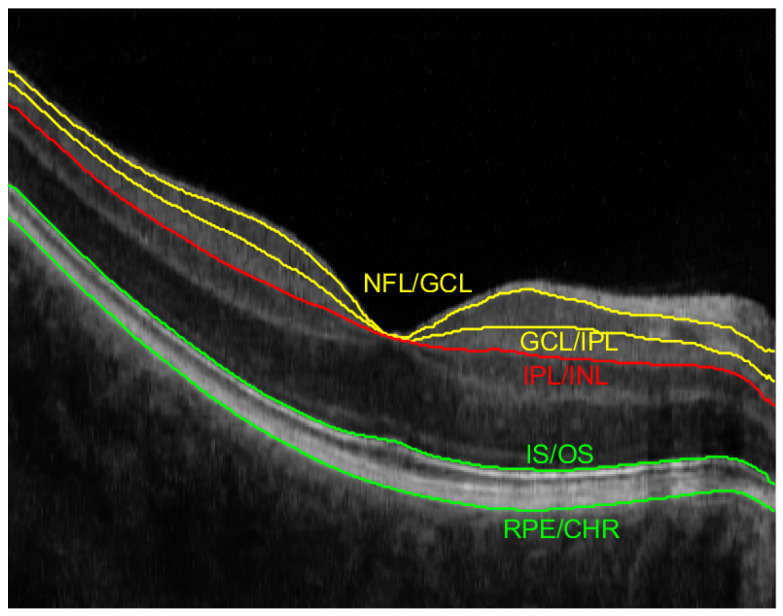
Example of OCT B-scan with annotated retina layers borders used for fundus reconstruction (NFL/GCL, GCL/IPL, IPL/INL, IS/OS, and RPE/CHR).

**Figure 3 sensors-23-01870-f003:**
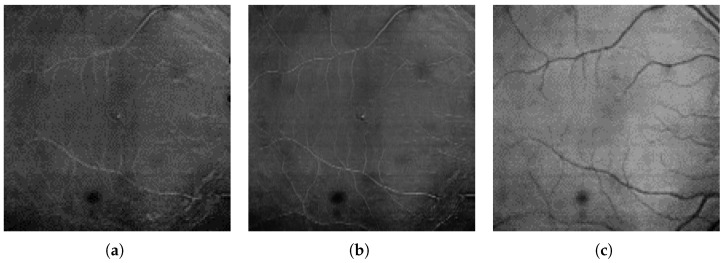
Example of projections obtained from selected retina layers: (**a**) GCL layer projection, (**b**) GCL+IPL layer projection, (**c**) OS+RPE layer projection.

**Figure 4 sensors-23-01870-f004:**
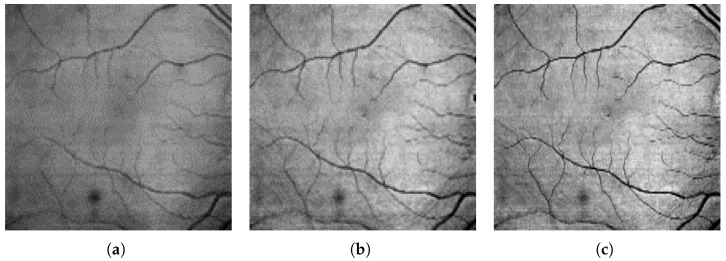
Example of resulting fundus reconstruction images using various retina regions: (**a**) P1—reconstruction from OS+RPE layers, (**b**) P2—reconstruction from GCL and OS+RPE layers, (**c**) P3—fundus reconstruction from GCL+IPL and OS+RPE layers.

**Figure 5 sensors-23-01870-f005:**
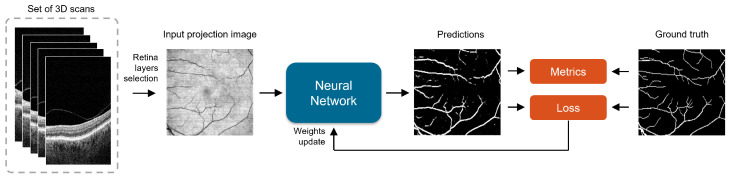
The general overflow of the proposed approach.

**Figure 6 sensors-23-01870-f006:**
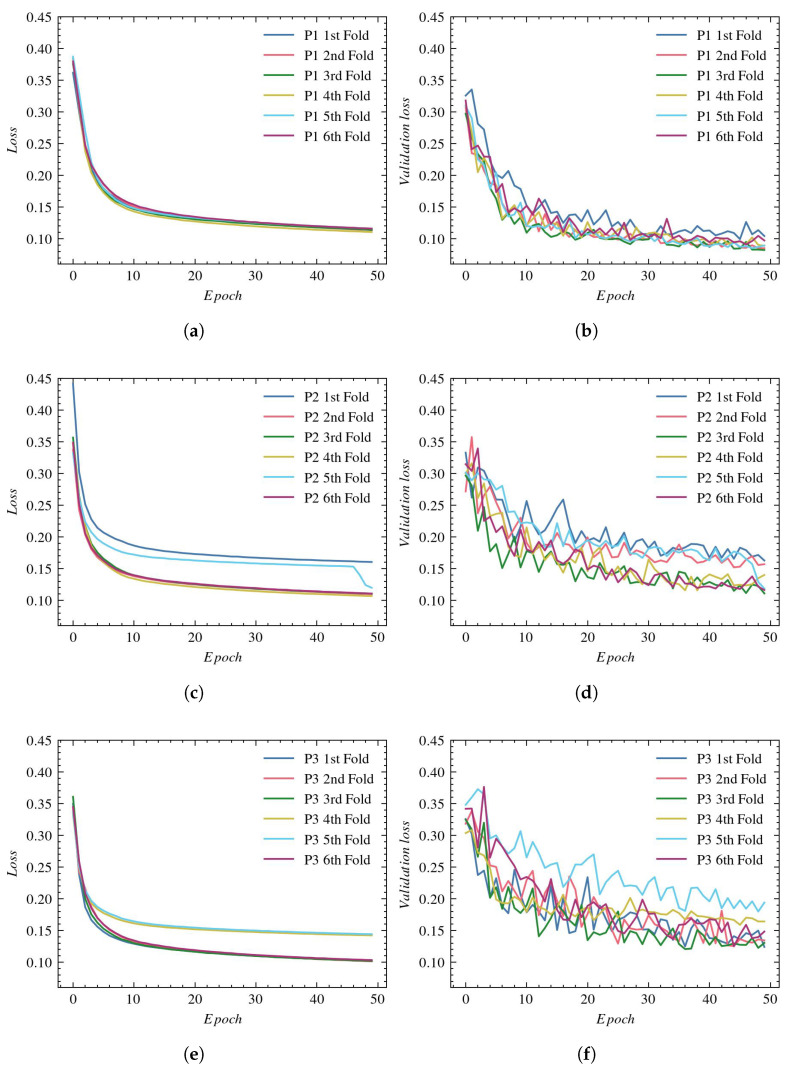
Loss plots for UNet architecture: (**a**) training using P1, (**b**) validation using P1, (**c**) training using P2, (**d**) validation using P2, (**e**) training using P3, and (**f**) validation using P3.

**Figure 7 sensors-23-01870-f007:**
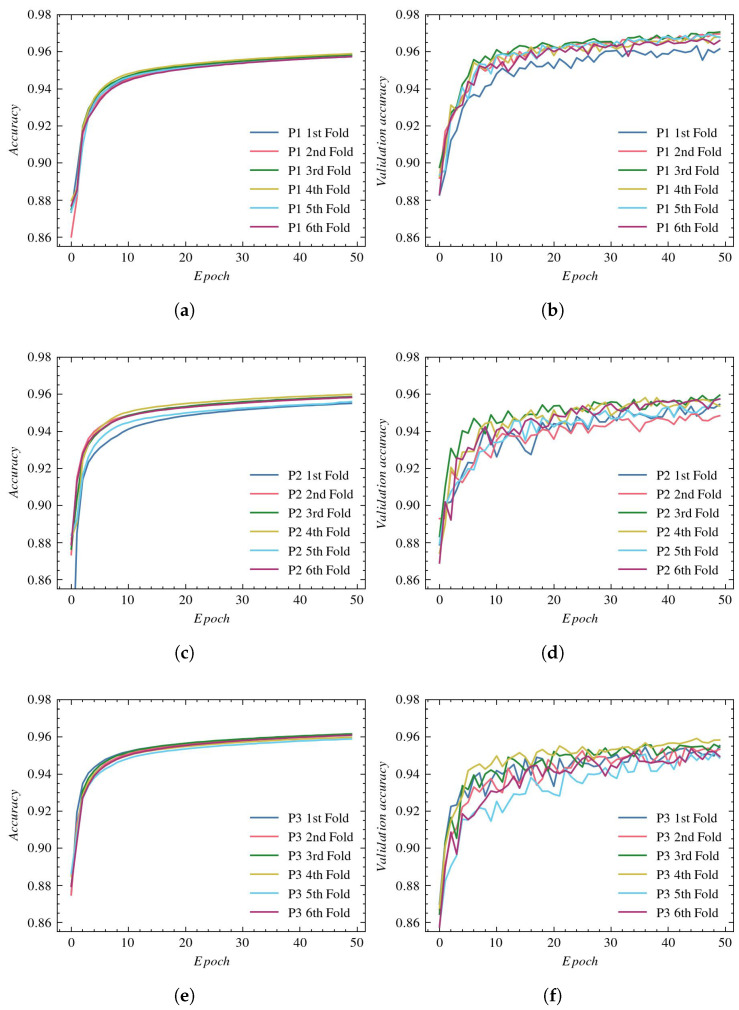
Accuracy plots for UNet architecture: (**a**) training using P1, (**b**) validation using P1, (**c**) training using P2, (**d**) validation using P2, (**e**) training using P3, and (**f**) validation using P3.

**Figure 8 sensors-23-01870-f008:**
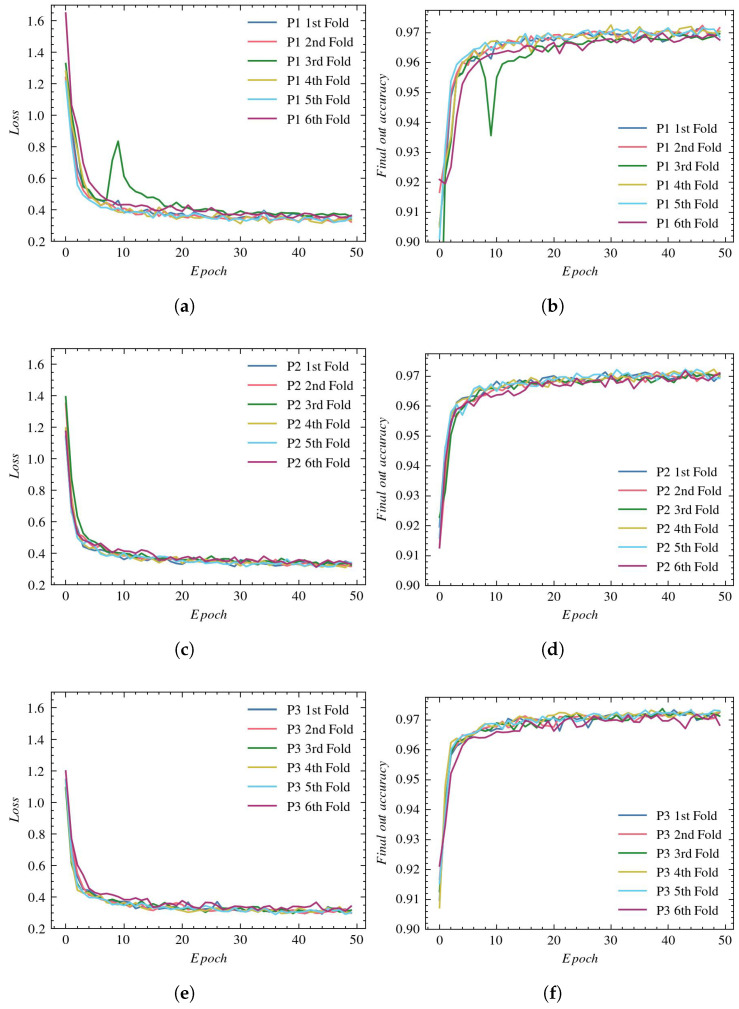
Loss and accuracy plots for IterNet architecture: (**a**) loss using P1, (**b**) accuracy using P1, (**c**) loss using P2, (**d**) accuracy using P2, (**e**) loss using P3, and (**f**) accuracy using P3.

**Figure 9 sensors-23-01870-f009:**
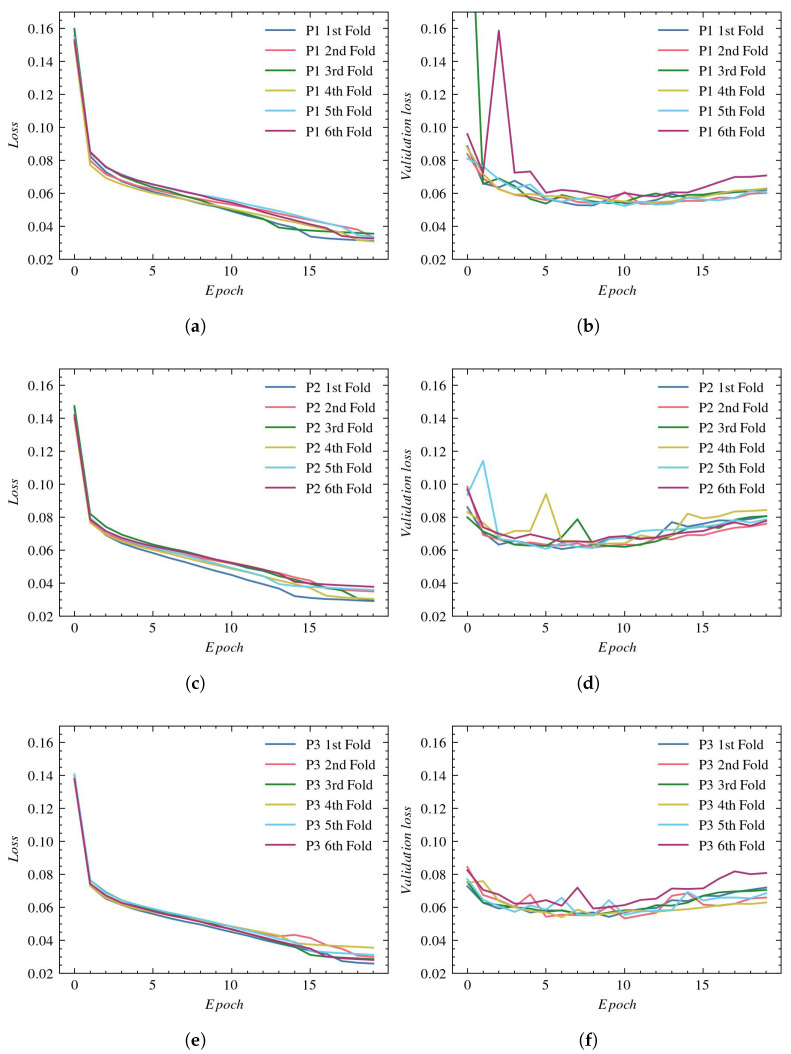
Loss plots for BCDU-Net architecture: (**a**) training using P1, (**b**) validation using P1, (**c**) training using P2, (**d**) validation using P2, (**e**) training using P3, and (**f**) validation using P3.

**Figure 10 sensors-23-01870-f010:**
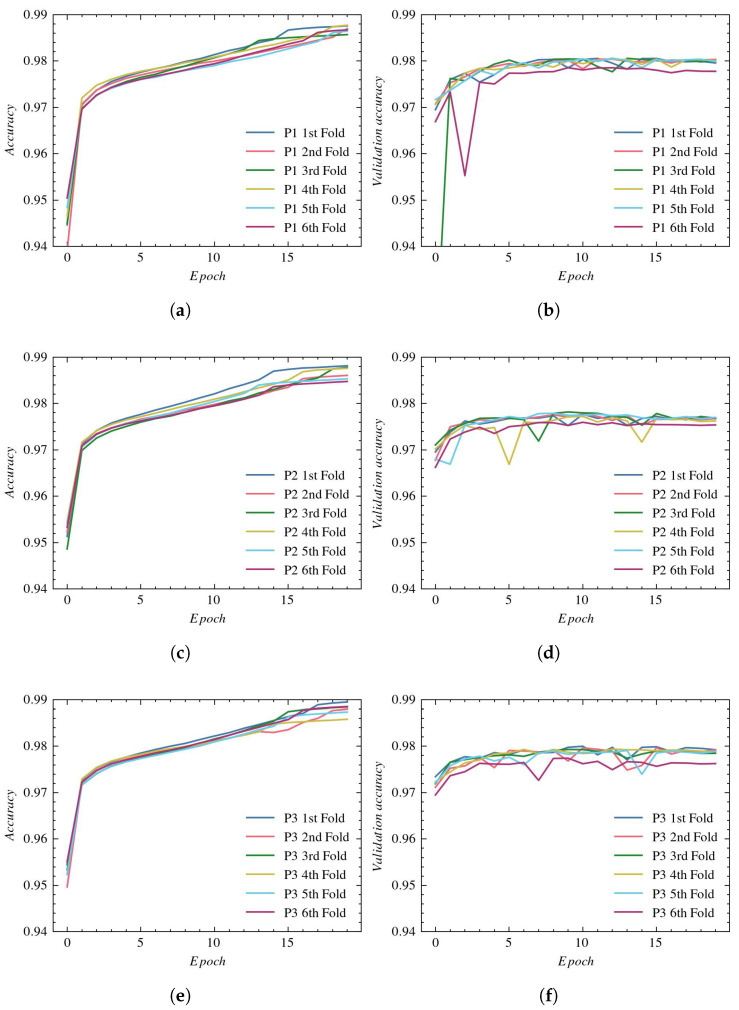
Accuracy plots for BCDU-Net architecture: (**a**) training using P1, (**b**) validation using P1, (**c**) training using P2, (**d**) validation using P2, (**e**) training using P3, and (**f**) validation using P3.

**Figure 11 sensors-23-01870-f011:**
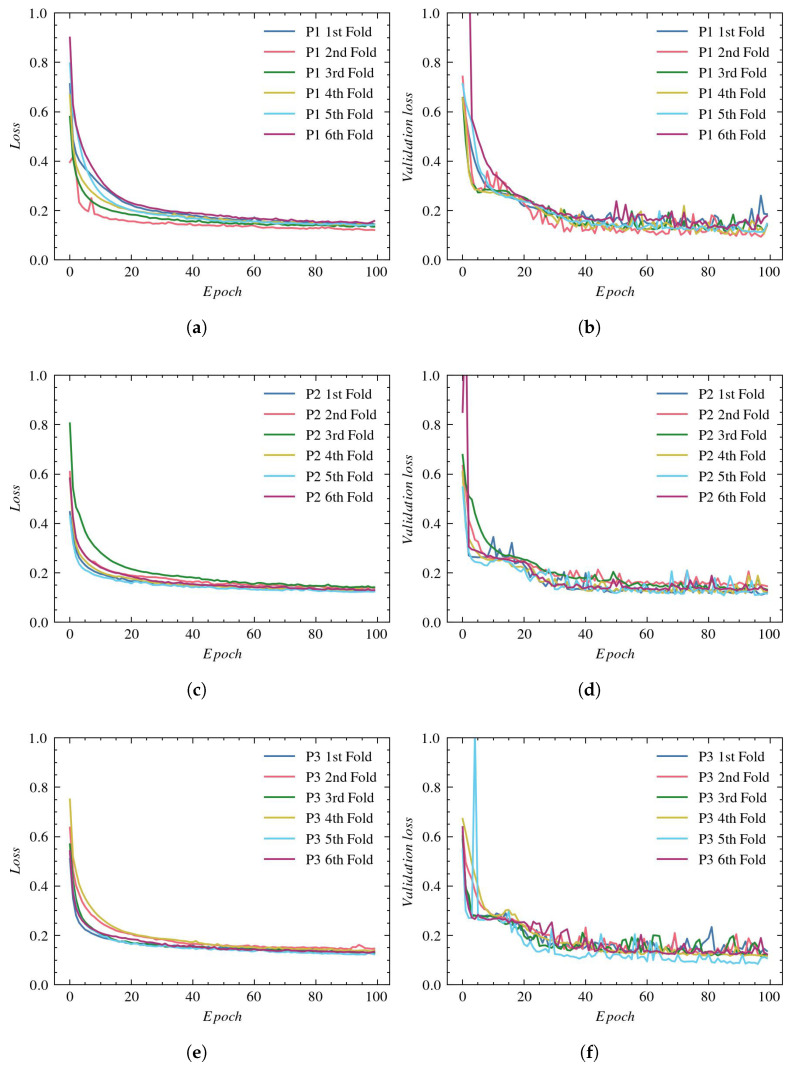
Loss plots for SA-UNet architecture: (**a**) training using P1, (**b**) validation using P1, (**c**) training using P2, (**d**) validation using P2, (**e**) training using P3, and (**f**) validation using P3.

**Figure 12 sensors-23-01870-f012:**
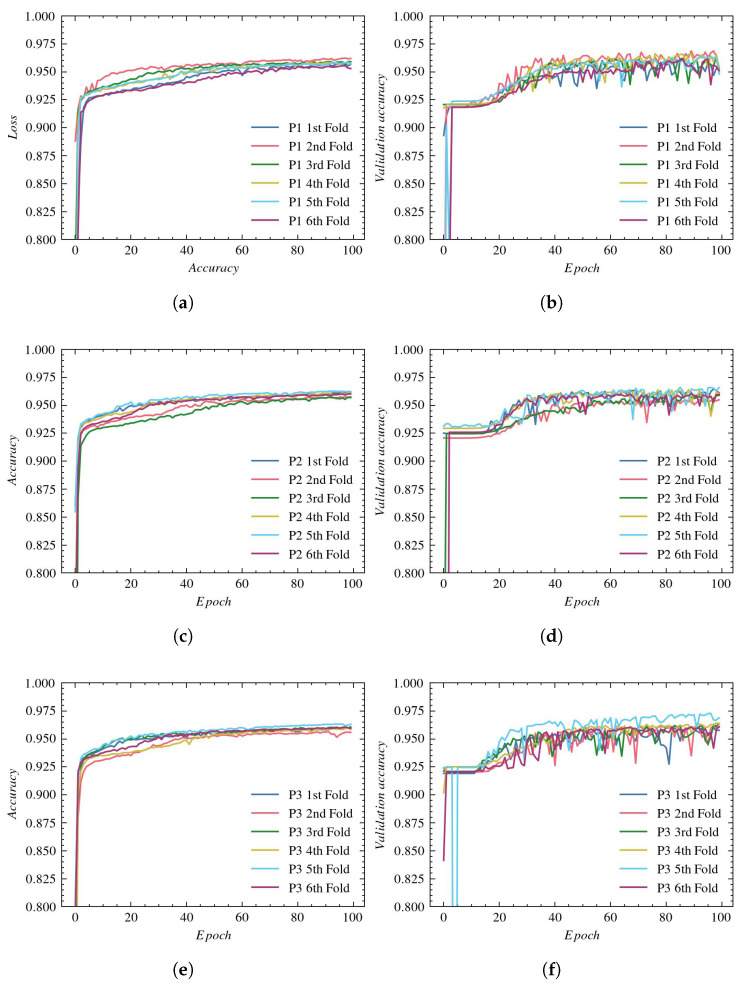
Accuracy plots for SA-UNet architecture: (**a**) training using P1, (**b**) validation using P1, (**c**) training using P2, (**d**) validation using P2, (**e**) training using P3, (**f**) validation using P3.

**Figure 13 sensors-23-01870-f013:**
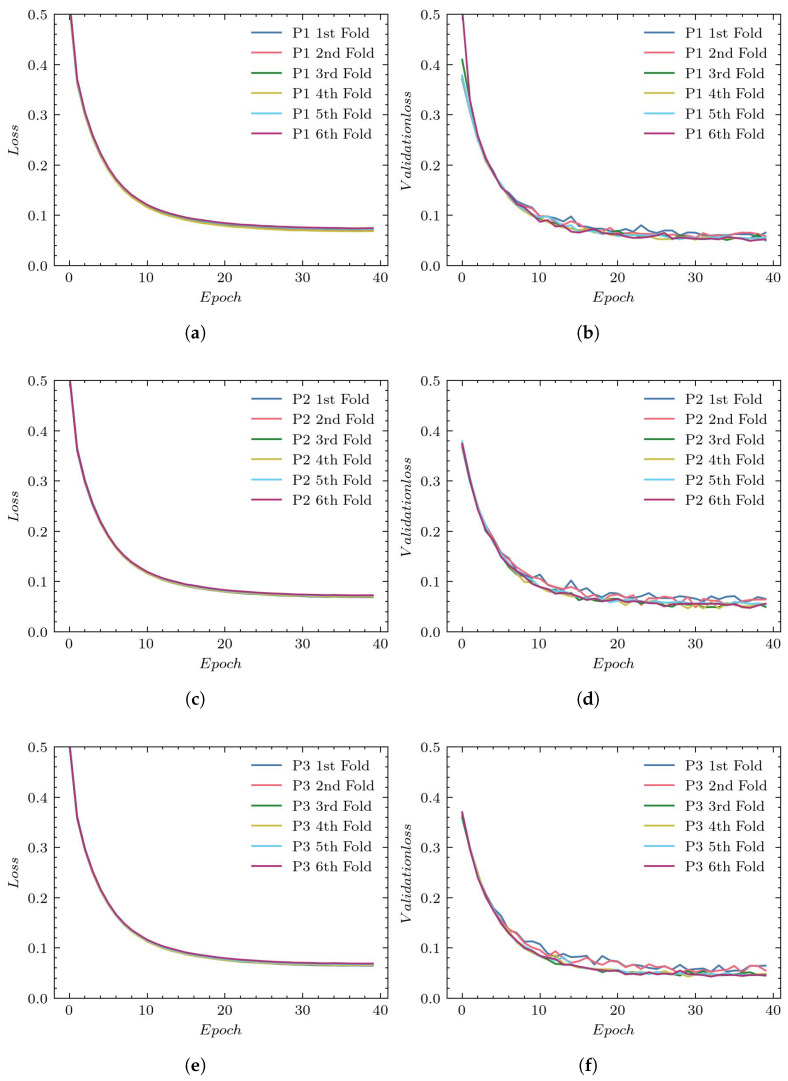
Loss plots for FR-UNet architecture: (**a**) training using P1, (**b**) validation using P1, (**c**) training using P2, (**d**) validation using P2, (**e**) training using P3, and (**f**) validation using P3.

**Figure 14 sensors-23-01870-f014:**
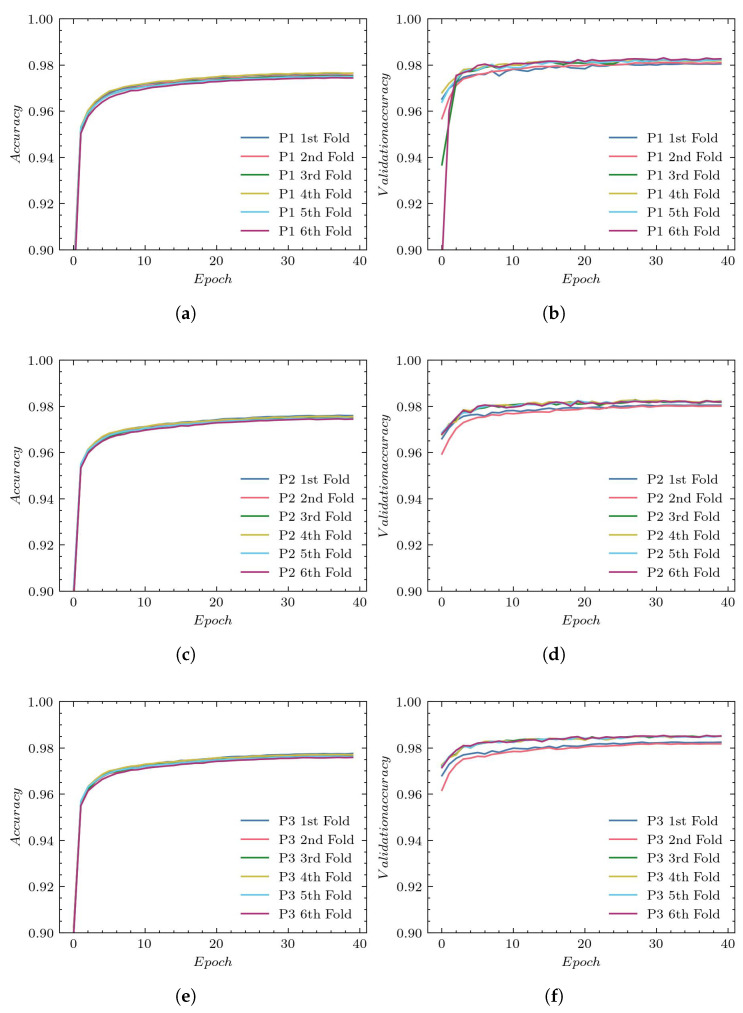
Accuracy plots for FR-UNet architecture: (**a**) training using P1, (**b**) validation using P1, (**c**) training using P2, (**d**) validation using P2, (**e**) training using P3, and (**f**) validation using P3.

**Figure 15 sensors-23-01870-f015:**
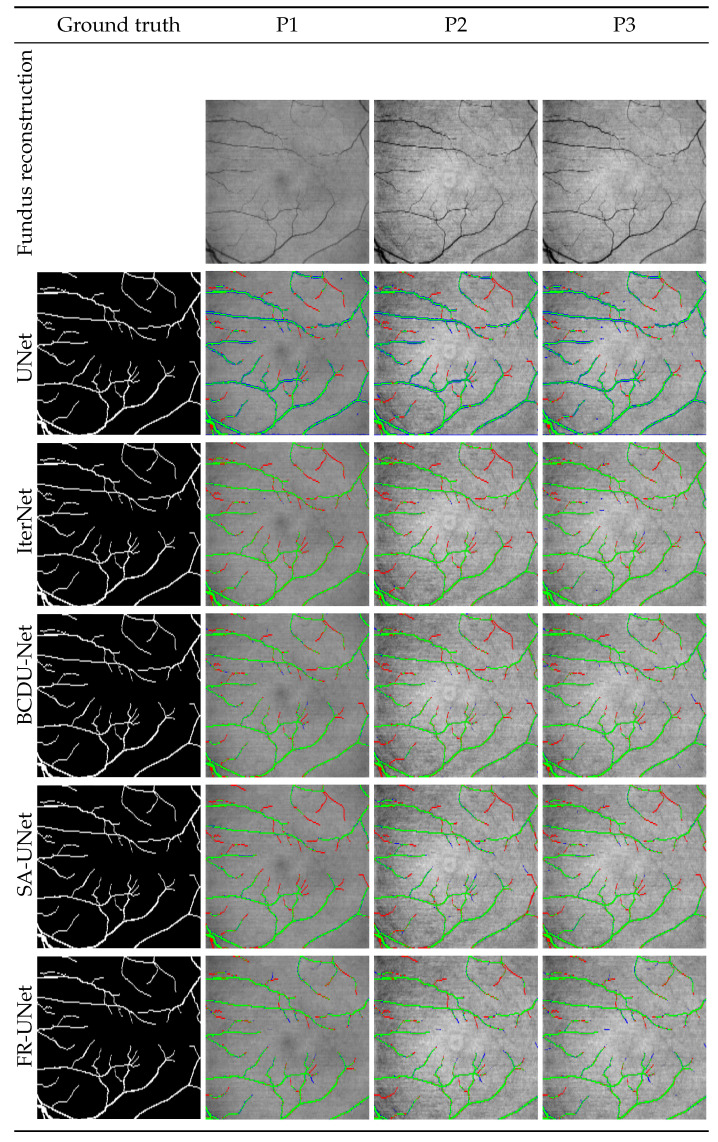
Example fundus reconstructions with ground truth and corresponding segmentation results for all neural networks.

**Table 1 sensors-23-01870-t001:** Parameters of the training setup.

Parameter	Network Architecture				
	**UNet**	**IterNet**	**BCDU-Net**	**SA-UNet**	**FR-UNet**
Number of epochs	50	50	20	100	40
Batch size	32	32	8	8	24
Optimizer	SDG lr = $1\times 10^{-2}$ momentum = 0	Adam lr = $1\times 10^{-3}$ $\beta_{1}$ = 0.9 $\beta_{2}$ = 0.999	Adam lr = $1\times 10^{-3}$ $\beta_{1}$ = 0.9 $\beta_{2}$ = 0.999	Adam lr = $1\times 10^{-3}$ $\beta_{1}$ = 0.9 $\beta_{2}$ = 0.999	Adam lr = $1\times 10^{-4}$ $\beta_{1}$ = 0.9 $\beta_{2}$ = 0.999
Loss function	Categorical cross-entropy	Binary cross-entropy	Binary cross-entropy	Binary cross-entropy	Binary cross-entropy
Dataset	20,000 patches 48×48 px	2000 patches 48×48 px generated every epoch	5000 patches 48×48 px	180 images 384×384 px	5120 patches 48×48 px
Training/Validation split	90%/10%	100%/0%	80%/20%	80%/20%	90%/10%

**Table 2 sensors-23-01870-t002:** Comparison of neural network models.

Method		Accuracy	Sensitivity	Specificity	Precision	F1-Score	AUC
Shadowgraph $s_{1}$ [15]		**0.932**	**0.511**	0.969	0.584	**0.538**	**0.749**
Shadowgraph $s_{2}$ [15]		0.904	0.326	0.954	0.389	0.350	0.637
Shadowgraph $s_{3}$ [15]		0.931	0.389	**0.979**	**0.612**	0.470	0.667
Morphological filtering—BML [22]		**0.972**	0.791	**0.986**	**0.839**	**0.816**	n/a
Morphological filtering—BDD [22]		0.924	**0.811**	0.932	0.508	0.625	n/a
Morphological filtering—BH [22]		0.907	0.656	0.930	0.450	0.534	n/a
**UNet**	P1	0.946	**0.862**	0.953	0.618	**0.718**	**0.974**
	P2	0.923	0.824	0.931	0.560	0.648	0.957
	P3	**0.948**	0.796	**0.962**	**0.648**	0.703	**0.974**
**IterNet**	P1	0.976	0.765	**0.993 ***	**0.910 ***	0.828	0.987
	P2	0.975	0.798	0.990	0.872	0.832	0.984
	P3	**0.977**	**0.835**	0.989	0.870	**0.851**	**0.990**
**BCDU-Net**	P1	**0.977**	0.815	**0.991**	**0.887**	0.848	0.989
	P2	0.975	0.796	0.990	0.880	0.834	0.988
	P3	**0.977**	**0.828**	0.990	0.874	**0.849**	**0.990**
**SA-UNet**	P1	0.974	0.794	0.989	0.868	0.828	0.977
	P2	0.973	0.782	0.989	0.864	0.816	0.979
	P3	**0.975**	**0.807**	**0.990**	**0.870**	**0.836**	**0.983**
**FR-UNet without DTI**	P1	0.977	0.851	**0.988**	**0.855**	0.851	0.989
	P2	0.976	0.842	**0.988**	0.853	0.845	0.989
	P3	**0.978 ***	**0.862**	**0.988**	**0.855**	**0.857 ***	**0.991 ***
**FR-UNet with DTI**	P1	**0.973**	0.903	0.978	0.783	0.837	n/a
	P2	0.972	0.895	**0.979**	0.783	0.833	n/a
	P3	**0.973**	**0.912 ***	**0.979**	**0.785**	**0.843**	n/a

## Data Availability

The reconstructed fundus images from CAVRI-C dataset used within this study can be found at http://dsp.org.pl/CAVRI_Database/191/ (accessed on 27 December 2022). Our implementation of utilized neural networks for cross-validation and compatibility with reconstructed fundus images can be found at the following links: UNet—https://github.com/przemyslaw-zaradzki/retina-unet_MSc (accessed on 27 December 2022). IterNet—https://github.com/przemyslaw-zaradzki/IterNet_MSc (accessed on 27 December 2022). BCDU-Net—https://github.com/przemyslaw-zaradzki/BCDU-Net_MSc (accessed on 27 December 2022). SA-UNet—https://github.com/przemyslaw-zaradzki/SA-UNet (accessed on 27 December 2022). FR-UNet—https://github.com/przemyslaw-zaradzki/FR-UNet (accessed on 27 December 2022).

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
