# Peer review of "Neural Networks Application for Accurate Retina Vessel Segmentation from OCT Fundus Reconstruction"

_sensors, 2023, doi:10.3390/s23041870_

Round 1
Reviewer 1 Report
I have the following concerns:
1, Traditional methods should also be tested, compared and discussed.
2, The limitations of all the compared methods should be summarized.
3, Why the F1-score is so low? Explanation is required.
4, The segmentation results and the ground truths should be superimposed on the original images for clear comparisons.
Author Response
Dear Editor,
Dear Reviewers,
We would like to thank the Reviewers for their time to read our manuscript and submit comments. Our answers are provided below. Changes to the text of the article are marked in blue.
We hope that our answers and changes to the text of the manuscript as suggested by the Reviewers are sufficient.
Yours sincerely,
Authors
- Traditional methods should also be tested, compared and discussed.
We thank the Reviewer for this comment. As suggested by the Reviewer, “Table 2. Comparison of neural network models” was supplemented with traditional methods, i.e., shadowgraph techniques and morphological operations.
Traditional methods are discussed in the section Related Work. It can be observed that the results that can be found in the cited papers are lower than the proposed solution. The table below summarizes the results presented by the authors in the cited publications and related to ours. It should be noted that in the related papers the authors use different quality indicators.
|
Method |
Results in paper |
Our results |
|
kNN classifier [16] |
AUC=0.970 |
AUC>0.974 in all cases (max. AUC=0.990) |
|
Multimodal (fundus + OCT) [1] |
AUC=0.900 (for NCO Entire Region) |
AUC>0.974 in all cases for P3 (max. AUC=0.990) |
|
Shadowgraph [15] |
Max. F1-score=0.538
Max. AUC=0.749 |
F1-score > 0.828 for IterNet, BCDUNet, FR-UNet (max.0.857 for FR-UNet) AUC>0.974 in all cases for P3 (max. AUC=0.990) |
|
Morphological operations BML [20] |
Max. F1-score=0.816 |
F1-score > 0.828 for IterNet, BCDUNet, FR-UNet (max.0.857 for FR-UNet) |
In addition, related solutions in some cases use multimodal data (fundus + OCT) [1], while our solutions are based solely on OCT. The paper [16] cited in Related Work does not contain a detailed description of the data processing procedure. For this reason, in Table 2 in our manuscript we have focused only on complete results for which software is available.
Additionally, the current version of the manuscript has been supplemented with an examination of the segmentation process using a new (Sept. 2022) neural network described in the paper:
- Liu et al., "Full-Resolution Network and Dual-Threshold Iteration for Retinal Vessel and Coronary Angiograph Segmentation," in IEEE Journal of Biomedical and Health Informatics, vol. 26, no. 9, pp. 4623-4634, Sept. 2022, doi: 10.1109/JBHI.2022.3188710.
2. The limitations of all the compared methods should be summarized.
The segmentation images show the problem of fragmented or lack of proper segmentation for the thinner vasculature. The main cause of this problem is the lower contrast between the thin vessels and the surrounding tissue, as well as low resolution of the image (the thin vessels have a width of 1 px).
The research was carried out for a specially developed database, including ground truth prepared by ophthalmology specialists. The effectiveness of the proposed and tested solutions is limited by the size of the database, the specificity of the scanning area (macula area) and the correctness of manual segmentation.
An important element of the proper working of the proposed solutions is also the precise segmentation of the retinal layers in OCT B-scans. These limitations have been included in the text of the paper.
- Why the F1-score is so low? Explanation is required.
The F1-score metric reaches slightly lower values, especially in the case of the UNet network (0.718, 0.648, 0.703). As we can see in the illustration in the attachment, the blood vessels marked by the algorithm (UNet in this case) with green (TP), blue (FP) and red (FN) colors are wider than the manual marking. This causes the F1-score metric (which does not use TN) to be lower.
The obtained F1-score results for fundus reconstruction are comparable to the results obtained by neural networks when segmenting vessels in color fundus images. For example, the authors [31] obtained F1-scores of 0.8316 in the case of color fundus images, which is similar to our OCT-based segmentation results for IterNet, BSDUNet, SA-UNet and FR-UNet (values are in the range from 0.816 to 0.857).
4, The segmentation results and the ground truths should be superimposed on the original images for clear comparisons.
According to the Reviewer's suggestion, the figure with sample segmentation results was changed.
Currently, Figure 15 contains comparisons where green colour represents a true positive pixel classifications, red represents a false negative, and blue represents a false positive.

Reviewer 2 Report
1. The working environment (i.e., software and hardware) and the experimental configurations (i.e., settings) should be added to a table.
2. The authors should conduct more experiments using different configurations.
3. The authors should provide an overall flowchart of the proposed approach.
4. The authors should add the equations of the used evaluation metrics, such as specificity, precision, sensitivity, F1-score, and accuracy.
Author Response
Dear Editor,
Dear Reviewers,
We would like to thank the Reviewers for their time to read our manuscript and submit comments. Our answers are provided below. Changes to the text of the article are marked in blue.
We hope that our answers and changes to the text of the manuscript as suggested by the Reviewers are sufficient.
Yours sincerely,
Authors
- The working environment (i.e., software and hardware) and the experimental configurations (i.e., settings) should be added to a table.
According to the reviewer's suggestion, Table 1 was prepared, in which the parameters of the tested neural networks were summarized.
- The authors should conduct more experiments using different configurations.
The results presented in Section 4 were carried out for the best-chosen parameters, in addition, the reader has access to the software in order to independently run the tests.
Our implementation of utilized neural networks for cross-validation and compatibility with reconstructed fundus images can be found at the following links:
- UNet – https://github.com/przemyslaw-zaradzki/retina-unet_MSc
- IterNet – https://github.com/przemyslaw-zaradzki/IterNet_MSc
- BCDUNet – https://github.com/przemyslaw-zaradzki/BCDU-Net_MSc
- SA-UNet – https://github.com/przemyslaw-zaradzki/SA-UNet
- FR-UNet – https://github.com/przemyslaw-zaradzki/FR-UNet
The current version of the manuscript has been supplemented with an examination of the segmentation process using a new neural network – FR-UNet (Sept. 2022) described in the article [31]:
- Liu et al., "Full-Resolution Network and Dual-Threshold Iteration for Retinal Vessel and Coronary Angiograph Segmentation," in IEEE Journal of Biomedical and Health Informatics, vol. 26, no. 9, pp. 4623-4634, Sept. 2022, doi: 10.1109/JBHI.2022.3188710.
- The authors should provide an overall flowchart of the proposed approach.
As suggested by the Reviewer, a flowchart was prepared, which is shown in Fig. 5.
- The authors should add the equations of the used evaluation metrics, such as specificity, precision, sensitivity, F1-score, and accuracy.
According to the Reviewer's suggestion, equations for determining accuracy, specificity, precision, sensitivity and F1-score were added.

Reviewer 3 Report
In this study, the authors examined four neural network architectures to segment vessels in fundus images reconstructed from 3D OCT scan data. The study is interesting and very well presented. I have only two minor comments:
1. Each figures should be placed directly after (not before) it was mentioned for the first time. Also, figure 10 belongs to subsection 4.6., however, it was placed above the title of the subsection.
2. Most of the conclusion is a discussion of the results. These paragraphs could be placed under the section (results and discussion), while the conclusion should be rewritten to be concise and to the point.
Author Response
Dear Editor,
Dear Reviewers,
We would like to thank the Reviewers for their time to read our manuscript and submit comments. Our answers are provided below. Changes to the text of the article are marked in blue.
We hope that our answers and changes to the text of the manuscript as suggested by the Reviewers are sufficient.
Yours sincerely,
Authors
In this study, the authors examined four neural network architectures to segment vessels in fundus images reconstructed from 3D OCT scan data. The study is interesting and very well presented. I have only two minor comments:
- Each figure should be placed directly after (not before) it was mentioned for the first time. Also, figure 10 belongs to subsection 4.6., however, it was placed above the title of the subsection.
We thank the Reviewer for pointing out the correct occurrence of figures in the manuscript. Currently, each of the figures appears after they are indicated in the text.
- Most of the conclusion is a discussion of the results. These paragraphs could be placed under the section (results and discussion), while the conclusion should be rewritten to be concise and to the point.
As suggested by the Reviewer, some paragraphs were moved to the previous section, while the conclusion was reformulated.

Round 2
Reviewer 1 Report
-
The introduction does not provide sufficient background. For instance, dual-threshold is used in [1], Is dual-threshold good enough? it should be compared with recent thresholding methods such as [2].
[1], Liu et al., "Full-Resolution Network and Dual-Threshold Iteration for Retinal Vessel and Coronary Angiograph Segmentation," in IEEE Journal of Biomedical and Health Informatics, vol. 26, no. 9, pp. 4623-4634, Sept. 2022, doi: 10.1109/JBHI.2022.3188710.
Author Response
Dear Reviewer,
We would like to thank the Reviewer for his time to read our manuscript and submit comments. Our answer is provided below. Changes to the text of the article are marked in blue.
We hope that our answers and changes to the text of the manuscript as suggested by the Reviewer are sufficient.
Yours sincerely,
Authors
Reviewer 1 – Review Report (Round 2)
The introduction does not provide sufficient background. For instance, dual-threshold is used in [1], Is dual-threshold good enough? it should be compared with recent thresholding methods such as [2].
[1], Liu et al., "Full-Resolution Network and Dual-Threshold Iteration for Retinal Vessel and Coronary Angiograph Segmentation," in IEEE Journal of Biomedical and Health Informatics, vol. 26, no. 9, pp. 4623-4634, Sept. 2022, doi: 10.1109/JBHI.2022.3188710.
[2], Z.Z. Wang,"Automatic localization and segmentation of the ventricle in magnetic resonance images," IEEE Transactions on Circuits and Systems for Video Technology, 31(2),621-631, 2021.
We thank the Reviewer for pointing out the above article [2] and the importance of thresholding. This article is now cited in section 1.1.2. A paragraph on thresholding has been added.
The double-threshold iterative (DTI) solution used with FR-UNet (Liu2022 article) was validated against our database and the results are added to the “Table 2. Comparison of neural network models”. It can be observed that the use of DTI only improves the Sensitivity score, but significantly reduces the Precision. Below we present the results for P1, P2 and P3 reconstructions with and without DTI.
In addition, tests were carried out for SDD (slope difference distribution) solution described in [2]. We used the software available at https://www.mathworks.com/matlabcentral/fileexchange/62574-image-segmentation-methods-comparison-with-mri Standard parameter settings (suggested in the software) were used. It can be observed that for our database (fundus reconstruction based on OCT images) the F1-score, Precision or Sensitivity results using SDD are low. This is probably due to the specificity of OCT images, which have different properties than MRI images, for which SSD works effectively.
P1 with DTI:
ROC: n/a
F1: 0.837
ACCURACY: 0.973
SENSITIVITY: 0.903
SPECIFICITY: 0.978
PRECISION: 0.783
P1 without DTI:
ROC: 0.989
F1: 0.851
ACCURACY: 0.977
SENSITIVITY: 0.851
SPECIFICITY: 0.988
PRECISION: 0.855
P1 with SDD:
ROC: n/a
F1: 0.613
ACCURACY: 0.973
SENSITIVITY: 0.65
SPECIFICITY: 0.984
PRECISION: 0.583
P2 with DTI:
ROC: n/a
F1: 0.833
ACCURACY: 0.972
SENSITIVITY: 0.895
SPECIFICITY: 0.979
PRECISION: 0.783
P2 without DTI:
ROC: 0.989
F1: 0.845
ACCURACY: 0.976
SENSITIVITY: 0.842
SPECIFICITY: 0.988
PRECISION: 0.853
P2 with SDD:
ROC: n/a
F1: 0.59
ACCURACY: 0.972
SENSITIVITY: 0.619
SPECIFICITY: 0.984
PRECISION: 0.567
P3 with DTI:
ROC: n/a
F1: 0.843
ACCURACY: 0.973
SENSITIVITY: 0.912
SPECIFICITY: 0.979
PRECISION: 0.785
P3 without DTI:
ROC: 0.991
F1: 0.857
ACCURACY: 0.978
SENSITIVITY: 0.862
SPECIFICITY: 0.988
PRECISION: 0.855
P3 with SDD:
ROC: n/a
F1: 0.596
ACCURACY: 0.972
SENSITIVITY: 0.627
SPECIFICITY: 0.984
PRECISION: 0.571
Once again, we would like to thank the Reviewer for valuable comments and tips that we will use in our next research related to the analysis of OCT images.
